# Optimization of Triterpene Saponins Mixture with Antiproliferative Activity

**Rodica Tatia** [1], **Christina Zalaru** [2,*], **Oana Craciunescu** [1], **Lucia Moldovan** [1], **Anca Oancea** [1] **and Ioan Calinescu** [3]

[1] Department of Cellular and Molecular Biology, National Institute of Research and Development for Biological Sciences, 296 Splaiul Independenţei, 060031 Bucharest, Romania; rodica_tatia@yahoo.com (R.T.); oana_craciunescu2009@yahoo.com (O.C.); moldovanlc@yahoo.com (L.M.); oancea.anca@gmail.com (A.O.)

[2] Department of Organic Chemistry, Biochemistry and Catalysis, Faculty of Chemistry, University of Bucharest, 90-92 Panduri Road, 50663 Bucharest, Romania

[3] Department of Bioresources and Polymer Science, Faculty of Applied Chemistry and Materials Science, 3 University POLITEHNICA of Bucharest, 1-7 Gh. Polizu Street, 011061 Bucharest, Romania; ioan.calinescu@upb.ro

\* Correspondence: chmzalaru@yahoo.com

**Abstract:** In this study, three of the saponins present in leaves of *Hedera helix* L., α-hederin, hederagenin, and hederacoside C were studied for their antiproliferative activity. The three saponins were analyzed in different concentrations by in vitro tests on normal fibroblasts cells and cervix ephitelial tumor cells. Determination of cytotoxicity and antitumor effects was performed using the MTT method. From the tested saponins, α-hederin was biocompatible in normal fibroblasts cells at concentrations between 2–10 µg/mL. Its antiproliferative activity was exerted in the concentration range of 10–400 µg/mL in cervix ephitelial tumor cells. Similarly, hederagenin presented antiproliferative activity at concentrations between 25–400 µg/mL. In turn, hederacoside C was shown to be noncytotoxic in normal fibroblasts and cervix ephitelial tumor cell culture at all the tested concentrations. The obtained experimental results were analyzed by "Mixture design", a specialized form of the response surface method (RSM) provided by the Design Expert 11 software, and the optimal composition of obtained saponins mixture was selected and verified in vitro for antiproliferative activity. The results showed that an optimal saponins mixture has the potential to be used in pharmacological applications.

**Keywords:** α-hederin; hederagenin; hederacoside C; cytotoxicity; antiproliferative activity; response surface method; mixture optimization

## 1. Introduction

Saponins are found in a variety of higher plants, and display a wide range of pharmacological activities, including expectorant, anti-inflammatory, vasoprotective, gastroprotective, antihelmintic, antileishmanial, antifungal, and antimicrobial properties [1–4]. Potential anticancer activity of saponins has been suggested by their antiproliferative, cytotoxic, cytostatic, pro-apoptotic, and anti-invasive effects [5–8].

Structurally, saponins are amphiphilic compounds composed of one or more hydrophilic sugar parts and a lipophilic steroidal or triterpenic part (sapogenin). Saponins can be classified into monodesmosidic, bidesmosidic, or polydesmosidic saponins, having one, two, or more sugar chains. A large structural variety can be found in nature, due to the presence of different sugars, sugar branchings, and sapogenins [9–11].

The interesting properties of saponins have led us to investigate the monodesmosidic saponin α-hederin, which has shown activities towards cancer cells [12]. α-Hederin, or Kalopanax saponin

A, was first isolated from the leaves of common ivy or *Hedera helix* L. (Araliaceae) [13,14], along with aglycone hederagenin, a triterpenic acid, by Van der Haar in 1912. Hederagenin is the aglycon part of numerous saponins found in *Hedera helix,*with the most prevalent being hederacoside C, a triterpene saponin with three sugar chains in its structure [15–17].

Triterpene saponins (hederacoside B, C, and D) and monodesmosides (α-, β-, δ-hederin, and hederagenin) were tested in vitro on four cell strains, mouse B16 tumor cells, 3T3 mouse normal fibroblasts, and HeLa human tumor cells, and presented about 5 times less antiproliferative activity compared to the reference compound strychnopentamine [18]. The most active compounds were α- and β-hederins, which were cytotoxic at concentrations greater than or equal to 10 μg/mL. Bidesmosidic hederacosides C, B, and D and hederagenin were inactive at concentrations up to 200 μg/mL. In another study, α-hederin extracted from *Clematis ganpiniana* was tested in two human breast cancer cell lines, MCF-7 and MDA-MB-231, inducing apoptosis in both types of cultures [19]. The researchers determined the strong inhibitory activity of α-hederin on breast tumor cells growth by apoptosis evaluation. Most antitumor therapies result in tumor inhibition by triggering cell apoptosis. Mitochondria play an important role in the survival of tumor cells; therefore, they are one of the main targets on which antitumor drugs work [20].

Apoptotic mitochondrial pathways have often been approached to highlight the activity of triterpene saponins in other human cancers, such as liver, gastric, esophagus, and colorectal cancer [21,22]. α-Hederin from *Nigella sativa* has been reported to induce apoptosis by mitochondrial perturbation in mouse P388 leukemia cells [23]. *In vitro* studies on the action mechanism of saponins had shown that saponins reacted and destroyed the lipids of the cell membrane [24,25]. In addition, α-hederin from *Hedera helix* L. presented anti-inflammatory and antileishmanial activities [26,27].

The aim of this study was to evaluate the in vitro cytotoxic and antiproliferative activity induced by the three triterpenic saponins, i.e., α-hederin, hederagenin, and hederacoside C, and to optimize a mixture of the three standard saponins for a maximal antiproliferative effect. For this purpose, the three standard saponins were tested for cytotoxicity in normal mouse fibroblasts (NCTC cells) and antiproliferative activity in human epithelial cervix tumor cells (Hep-2). The obtained experimental results were analyzed using the Response Surface Method (RSM) provided by the Design Expert 11 software (Version: 11.1.0.1 64 bit, Stat-Ease Inc., Minneapolis, MN, USA, 2018 [28], correlating the concentration of studied saponins with the antiproliferative activity. A mixture design model using Design Expert 11 was proposed and tested.

## 2. Materials and Methods

### 2.1. Materials

Standard saponins α-hederin, hederagenin, and hederacoside C at purity ≥98% (HPLC), and 3-(4,5-dimethyl-thiazol-2-yl)-2,5-diphenyltetrazolium bromide (MTT) and dimethyl sulfoxide (DMSO) were purchased from Sigma Aldrich Germany. NCTC clone L929 cell culture of mouse normal fibroblasts and Hep-2 tumor cell culture of human cervix epithelial cells were purchased from ECACC (Sigma-Aldrich, Merck Group, Darmstadt, Germany), together with Minimum Essential Medium (MEM), L-glutamine, and antibiotics penicillin, streptomycin, and neomycin. Fetal bovine serum (FBS) was provided by Biochrom (Cambridge, UK). Standard saponins are soluble in DMSO.

### 2.2. Methods

#### 2.2.1. Cell Culture

The cell cultures used for in vitro testing were cultivated in MEM, supplemented with 10% FBS, 2 mM L-glutamine and a mixture of 100 U/mL penicillin, 100 μg/mL streptomycin, and 500 μg/mL neomycin.

Fibroblasts were seeded at a density of $4 \times 10^4$ cells/mL for NCTC culture and $4 \times 10^4$ cells/ mL, while Hep-2 cells were seeded at a density of $6 \times 10^4$ cells/ mL in 96-well culture plates and incubated at 37°C in a humid atmosphere with 5% $CO_2$ for 24 h using a Bioquell biology security cabinet.

Standard saponins were solubilized in a small amount of DMSO and diluted in the culture medium to obtain stock solutions which were filtered through 0.22 μm fillters (Millipore Merck, Burlington, MA, USA). They were added tothe culture medium at concentrations of 2, 5, 10, 25, 50, 100, 200, 300, and 400 μg/mL. The compounds tested in NCTC culture were diluted in MEM supplemented with 10% FSB, while for Hep-2 tumor cell culture, a parallel testing was performed in two variants: in the first one, the samples were diluted in MEM supplemented with 10% FBS, and in the other one, the samples were diluted in FBS-free of MEM [8]. In addition, mixtures of saponins determined by mathematical modeling were also tested in the same conditions. The tested samples were added in triplicate to the wells of the culture plates. The culture plates were incubated under standard conditions (37 °C, 5% $CO_2$) for 24 h. Untreated cultured cells were used as control.

### 2.2.2. *In Vitro* Cytotoxicity Testing by MTT Assay

Cell viability after cell cultivation with samples was determined using theMTT colorimetric method in which tetrazolium salt reacted with mitochondrial dehydrogenases from the metabolic active cells reducing, with the formation of blue-violet formazan crystals which were insoluble in the culture medium [29]. For the cytotoxicity assay, the culture medium was replaced with MTT solution at a concentration of 50 μg/mL, followed by culture plate incubation for 3 h. After incubation, isopropanol was added to solubilize the formazan crystals that formed in viable cells by shaking the plates for 15 min on an orbital shaker. Determination of cell viability was performed after 24 h of treatment by optical density (OD) measurement of the colored solution at a wavelength of 570 nm. The measured OD is directly proportional to the number of viable cells present in the cell culture. The results were calculated using the following formula:

$$\% \ cell \ viability = (OD \ sample / OD \ control) \times 100\%$$

Control cells were considered to have 100% viability.

### 2.2.3. Mixture modeling using Design Expert 11 software

In a mixture experiment, the independent factors are proportions of different components of a blend, and the measured response is assumed to depend only on the relative proportions of the ingredients or components in the mixture [30].

The Response Surface Method (RSM) provided by Design Expert 11 software is a specialized statistical tool used to model the blending surface with some form of mathematical equation [28,31], so that:

1. Predictions of the response for any mixture or combination of the ingredients can be made empirically or statistically.
2. Some measure of the influence on the response of each component and in combination with other components can be obtained.
3. The optimization feature can be used to calculate the optimum composition for a mixture to maximize a certain effect.

### 2.2.4. Statistical Analysis

The results of cell culture experiments were expressed as mean value ± standard deviation (SD) of three independent samples (n = 3). Statistical analysis of the data was performed using one-tailed, paired Student's *t*-test (Excel, Office Excel 2007 software, Microsoft, Redmond, WA, USA) on each pair of interest. Differences were considered statistically significant at $p \leq 0.01$.

## 3. Results and Discussion

### 3.1. Influence of Studied Saponins on Cell Viability

3.1.1. Chemical Formula of Saponins

The structures of studied saponins are presented in Figure 1.

Hederagenin

Chemical formula $C_{30}H_{48}O_4$

Molar mass = 472.1g/mol

$\alpha$-Hederin

Chemical formula $C_{41}H_{67}O_{12}$

Molar mass = 751.4g/mol

Hederacoside C

Chemical formula $C_{59}H_{96}O_{26}$    Molar mass = 1221.38 g/mol

**Figure 1.** Saponins structures.

3.1.2. Cytotoxicity of Saponins in NCTC Cells

The results of the MTT assayfor the standard saponins in NCTC cells are presented in Figure 2.

The values indicated that the compound $\alpha$-hederin presented a high degree of biocompatibility in NCTC cells in the concentration range of 2–10 µg/mL (99–82% cell viability) at 24 h. It was slightly cytotoxic at 25 µg/mL (62% cell viability) and severely cytotoxic throughout the range of concentrations 50–400 µg/mL, when the cell viability decreased to 0.4%.

Hederagenin was biocompatible in NCTC cells from 2 to 50 µg/mL, had moderate cytotoxicity in the range of 100–300 µg/mL, with values of 73–56% cell viability, and was cytotoxic at 400 µg/mL, with only 49% cell viability at 24 h.

Hederacoside C was biocompatible at concentrations ranging between 2–300 µg/mL, with high cell proliferation (105–88% viability) at 24 h, compared to the control culture, and non cytotoxic (79% viability) at 400 µg/mL.

Comparing the cytotoxicity assay results of the three studied saponins in NCTC cells, it was concluded that α-hederin was biocompatible only up to 10 μg/mL, while in the concentration range of 25–400μg/mL, it was cytotoxic; hederagenin had moderate cytotoxicity between 100–300μg/mL concentrations, and increased cytotoxicity at 400μg/mL, while hederacoside C was biocompatible in the concentration range of 2–400μg/mL.

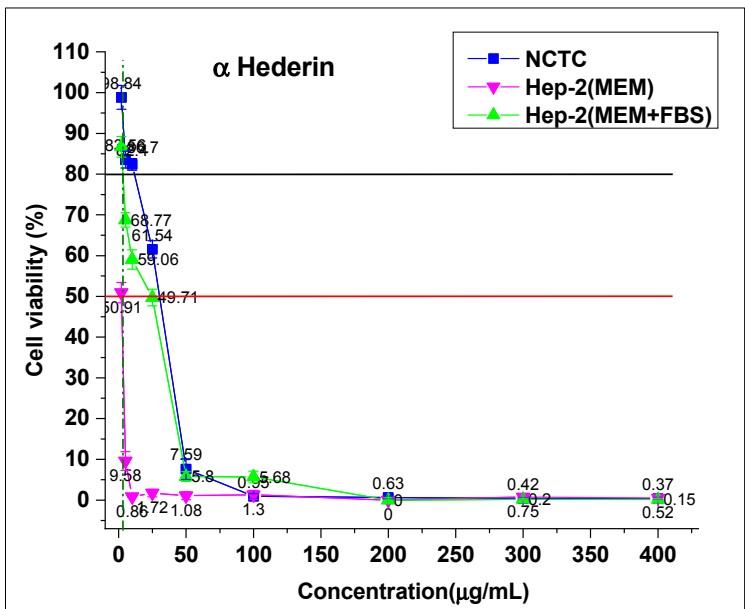

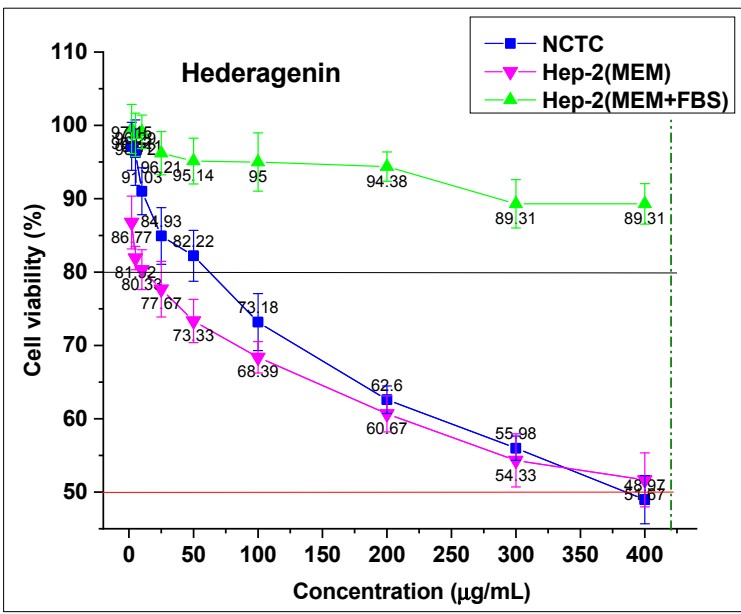

**Figure 2.** *Cont.*

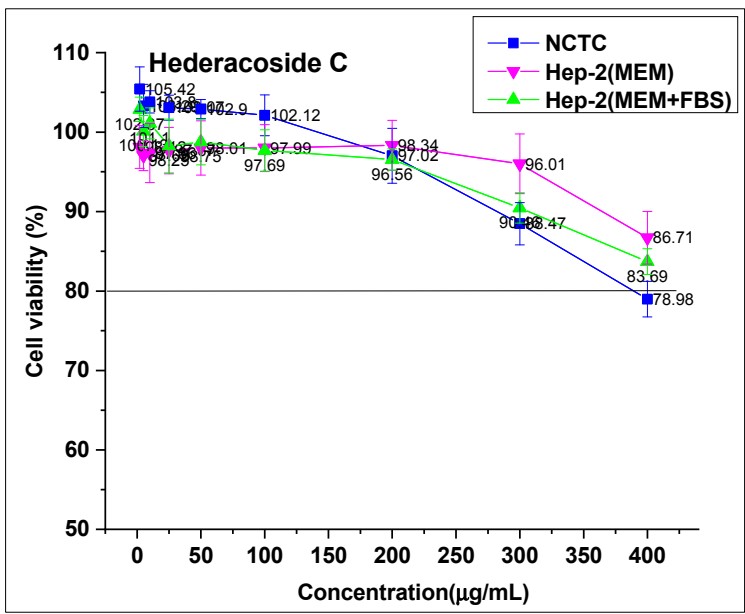

**Figure 2.** *In vitro* cytotoxicity of standard saponins (α-hederin, hederagenin and hederacoside C) tested in fibroblasts cells (NCTC) and tumoral cells (Hep-2) tested in MEM culture medium or MEM supplemented with FBS (MEM + FBS), determinated by MTT assay. The values represent mean ± SD (n = 3), $p < 0.01$.

### 3.1.3. Antiproliferative Activity of Saponins in Hep-2 Cells

The MTT testresults for the analyzed standard saponins in Hep-2 tumor cells indicated that α-hederin induced the most potent antitumoral effect (Figure 2). At a concentration of 2 μg/mL, α-hederin was biocompatible in NCTC cells (99% cell viability), while in Hep-2 tumor cells cultivated in medium, it was antiproliferative (51% viability) after 24 h of treatment.

The MTT assay results for hederagenin tested in Hep-2 cells cultivated in a FBS-free medium exerted a moderate cytotoxicity activity at 25–300 μg/mL and strong cytotoxicity at 400 μg/mL (52% cell viability). In cells cultivated in a medium with FBS, it was observed that hederagenin had no significant antitumor activity, with the viability of Hep-2 tumoral cells exceeding 80% within the entire range of concentrations (2–400 μg/mL).

Hederacoside C was biocompatible within the entire range of concentrations (2–400 μg/mL), with no significant antitumoral effect when cultivated in medium with or without FBS (Figure 2).

Our study confirmed that saponins tested in the medium containing FBS presented very low antiproliferative activity. Hederagenin in the medium with FBS did not present any cytotoxicity in Hep-2 tumor cells, compared to the experiment in medium without FBS.

The antitumor activity of standard saponins was also calculated as the inhibitory concentration of compounds that killed 50% of cells (IC$_{50}$, μg/mL). The results are presented in Table 1.

**Table 1.** Values of IC$_{50}$ determined for the saponins cultivated in Hep-2 cell. Culture cultivated in medium with and without fetal bovine serum (FBS).

| Standard | IC$_{50}$ (μg/mL) | |
|---|---|---|
| | **Hep-2 MEM** | **Hep-2 MEM + FBS** |
| α-Hederin | 2 | 25 |
| Hederagenin | 413 | >500 |
| Hederacoside C | >400 | >400 |

$\alpha$-Hederin presented the lowest value of IC$_{50}$ in Hep-2 cells in the medium without FBS (2 µg/mL) and with FBS (25 µg/mL), indicating the highest antiproliferative activity from the three tested saponins. In turn, hederagenin and hederacoside C presented low antiproliferative activity (IC$_{50}$>400 µg/mL).

Previous in vitro studies suggested that the chemical bonds between saponins and proteins from fetal calf serum (FCS) reduced their antitumor effect [8]. In our saponin antitumoral activity evaluations, the growth factor added in the culture medium was FBS, known to have a similar composition to FCS and to be rich in proteins such as albumin, globulin, and gamma globulin [32]. FBS also diminishedthe antitumor activity of saponins by protein binding. Thus, in our in vitro study, saponin samples were processed (individually and mixtures variants) in a medium supplemented, or not, with FBS.

Similar studies showed that the most active compounds in mouse B16 tumor cells and 3T3 mouse normal fibroblasts $\alpha$- and $\beta$-hederins were cytotoxic at concentrations greater than or equal to 10 µg/mL [19]. Other in vitro experiments on B16 mouse melanoma and HeLa human tumor cells cultivated in serum-free media showed that $\alpha$-hederin was toxic at concentrations less than 5 µg/mL after only 8 h of treatment [33]. $\alpha$-Hederin induced the vacuolization of the cytoplasm and the alteration of the membranes that caused cell death. However, the cytotoxicity of the phytocompound was reduced in the presence of fetal calf serum (FCS) or bovine serum albumin (BSA) in the culture medium due to interactions between $\alpha$-hederin and proteins.

In a study of hederagenin apoptosis effects and its possible mechanism of action in human colon cancer LoVo cells, the MTT assay showed significant inhibition of cell proliferation in a concentration- and time-dependent manner. The IC$_{50}$ was 1.39 µM at 4 h and 1.17 µM at 48 h of cultivation. The apoptosis percentage increased from 32% to 82% when the hederagenin concentration increased from 1 to 2 µM [34].

### 3.2. Mixture Modeling Using Design Expert 11 Software

In our study, the obtained cytotoxicity and antiproliferative experimental results were analyzed by RSM, provided by Design Expert 11, correlating the composition of the studied saponin mixtures with cytotoxic properties in order to optimize the saponin mixture with an antiproliferative effect.

The considered components and their selected concentration limits, estimated based on cytotoxicity assay results, are presented in the Table 2.

**Table 2.** Independent factors considered and their variation limits in µg/mL.

| Component | Name | Min. | Max. |
|:---:|:---:|:---:|:---:|
| **A** | $\alpha$-Hederin | 2 | 8 |
| **B** | Hederagenin | 100 | 498 |
| **C** | Hederacoside C | 200 | 598 |
| | | **Total=** | **700.00** |

For the experiment design, the following features of the experimental matrix were used:

- Model: Scheffe
- Lack of fit points: 5
- Replicate points: 3
- Additional model points: 0
- Additional center points: 0
- Blocks: 1
- Total runs: 14

The software generated 14 variants of mixture compositions for the tested saponins presented in Table 3.

**Table 3.** Experimental matrix of the mixture modeling.

| Run | Mixture Composition, µg/mL | | |
|---|---|---|---|
| | A: α-Hederin | B: Hederagenin | C: Hederacoside C |
| 1 | 8.000 | 381.029 | 310.971 |
| 2 | 3.824 | 100.000 | 596.176 |
| 3 | 2.000 | 279.100 | 418.900 |
| 4 | 3.514 | 496.486 | 200.000 |
| 5 | 8.000 | 139.800 | 552.200 |
| 6 | 2.000 | 343.630 | 354.370 |
| 7 | 2.000 | 188.104 | 509.896 |
| 8 | 5.305 | 301.292 | 393.403 |
| 9 | 3.824 | 100.000 | 596.176 |
| 10 | 5.305 | 301.292 | 393.403 |
| 11 | 2.000 | 424.557 | 273.443 |
| 12 | 8.000 | 462.150 | 229.850 |
| 13 | 8.000 | 230.898 | 461.102 |
| 14 | 5.305 | 301.292 | 393.403 |

### 3.3. In Vitro Cytotoxicity and Antiproliferative Activity of the Saponin Mixtures

Each of the 14 variants of saponin mixtures generated by Design Expert 11 software were tested in a similar way used for the cyotoxicity evaluation of standard saponins, with determination of cell viability in both normal and tumor cells. The obtained results are presented in Table 4.

**Table 4.** *In vitro* cytotoxicity results for saponin mixtures. (α-hederin, hederagenin and hederacoside C) tested in fibroblasts cells (NCTC) and tumoral cells (Hep-2), determined by MTT assay.

| Mixture Variants | Cell Viability (%) | | |
|---|---|---|---|
| | NCTC | Hep-2 (MEM) | Hep-2 (MEM + FBS) |
| 1 | 44 ± 2.04 | 4 ± 1.59 | 49 ± 2.82 |
| 2 | 77 ± 1.15 | 37 ± 1.24 | 77 ± 3.44 |
| 3 | 72 ± 3.59 | 60 ± 3.84 | 66 ± 3.59 |
| 4 | 63 ± 1.41 | 38 ± 3.68 | 69 ± 1.55 |
| 5 | 69 ± 1.82 | 2 ± 0.95 | 61 ± 3.00 |
| 6 | 73 ± 4.04 | 49 ± 3.31 | 64 ± 2.97 |
| 7 | 79 ± 2.58 | 57 ± 3.74 | 73 ± 3.98 |
| 8 | 60 ± 0.61 | 2 ± 1.24 | 63 ± 0.35 |
| 9 | 81 ± 0.31 | 36 ± 1.34 | 77 ± 3.08 |
| 10 | 65 ± 2.32 | 2 ± 0.41 | 67 ± 2.56 |
| 11 | 71 ± 2.49 | 63 ± 3.69 | 69 ± 3.35 |
| 12 | 50 ± 3.02 | 17 ± 0.31 | 44 ± 2.98 |
| 13 | 65 ± 1.31 | 2 ± 0.72 | 70 ± 2.28 |
| 14 | 63 ± 3.03 | 2 ± 2.11 | 62 ± 1.20 |
| Response range | 44 to 81 | 2 to 64 | 44 to 77 |
| Ratio of max to min | 1.84 | 39.25 | 1.73 |
| Transformation | none | Natural log | none |

The values represent mean ±SD (n = 3).; $p < 0.01$

From the values obtained for cell viability at 24 h of Hep-2 response in medium without FBS, it can be seenthat for the variation limits were very large and it was used as a means of transformation: ln (Y).

### 3.4. Statistical Analysis

Using analysis of variance, the software proposed the models presented in the Tables 5 and 6. For each model, the F-and *p*-values were determined. These values showed that the models were significant. Also, the lack of fit of the F-and *p*-values implied that lack of fit was not significant relative to the pure error. The *p*-values were determined for each coefficient, with values greater than 0.1000 indicating the model terms were not significant (Table 5).

**Table 5.** Model type, F-and *p*-values for the model, and for lack of fit and coefficient of regression.

| Response | Model Type | Model | | Lack of Fit | | $R^2$ |
|---|---|---|---|---|---|---|
| | | F-Value | *p*-Value | F-Value | *p*-Value | |
| NCTC | Quadratic model | 22.77 | 0.0002 | 2.79 | 0.2142 | 0.9343 |
| Hep-2 (MEM) | Cubic model | 457.91 | <0.0001 | 5.18 | 0.1072 | 0.9990 |
| Hep-2(MEM+FBS) | Special Quartic model | 52.48 | 0.0002 | 0.191 | 0.8355 | 0.9882 |

**Table 6.** The model coefficients and associated *p*-values.

| Model | Coefficients (Coded Equation) and *p*-Value | | | | | | | | | | |
|---|---|---|---|---|---|---|---|---|---|---|---|
| | A | B | C | AB | AC | BC | ABC | AB(A-B) | AC(A-C) | ABC² | A |
| NCTC | 18,786.6 | 70.9735 | 83.3454 | −20,732.2 | −19,596.5 | 15.8615 | - | - | - | - | 18,786.6 |
| *p*-values | <0.0001 | <0.0001 | <0.0001 | 0.5880 | 0.6085 | 0.1430 | - | - | - | - | <0.0001 |
| ln Hep-2 (MEM) | 5.65968 E+06 | 4.51276 | 3.99676 | 8.53632 E+06 | 8.552 E+06 | 1.0551 | 5.76906 E+06 | 2.87669 E+06 | 2.89269 E+06 | - | 5.65968 E+06 |
| *p*-values | <0.0001 | <0.0001 | <0.0001 | 0.0052 | 0.0052 | 0.3829 | 0.0052 | 0.0053 | 0.0052 | - | <0.0001 |
| Hep-2 (MEM + FBS) | −9343.46 | 76.5045 | 88.5913 | 7533.95 | 6852.24 | 71.2811 | - | - | - | 16,716.9 | −9343.46 |
| *p*-values | <0.0001 | <0.0001 | <0.0001 | 0.9173 | 0.9248 | 0.0152 | - | - | - | 0.0391 | <0.0001 |

Using the models, we were able todescribe the response surface for each variant. The charts were ternary, and because there was a large difference between the α-hederin variation domains and the other two components, we had to present the larger diagrams in the so called concentration zone (Figure 3).

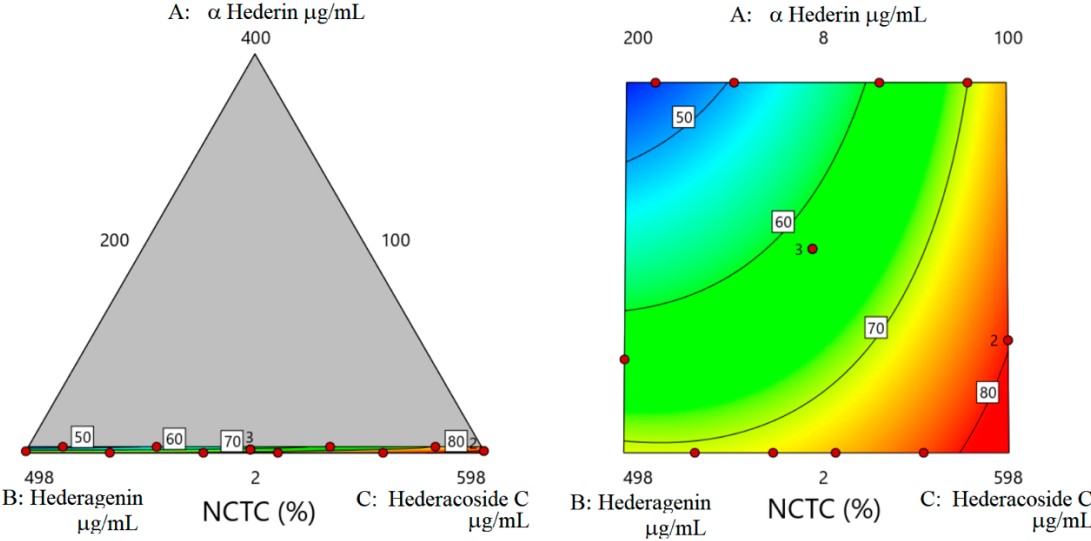

**Figure 3.** Response surface for cell viability of NCTC fibroblasts ternary and concentration zone.

From the analysis of the results presented in Figure 4, it may be seen that the viability of Hep-2 cells was lower in the presence of MEM without FBS than with FBS. Areas with cell viability below 50% were determined in particular by the concentration of α-hederin for Hep-2 (MEM), and for Hep-2 (MEM + FSB), the involved factors were more numerous and the area with viability below 50% was much more restricted (see also the coefficients of the model in Table 5).

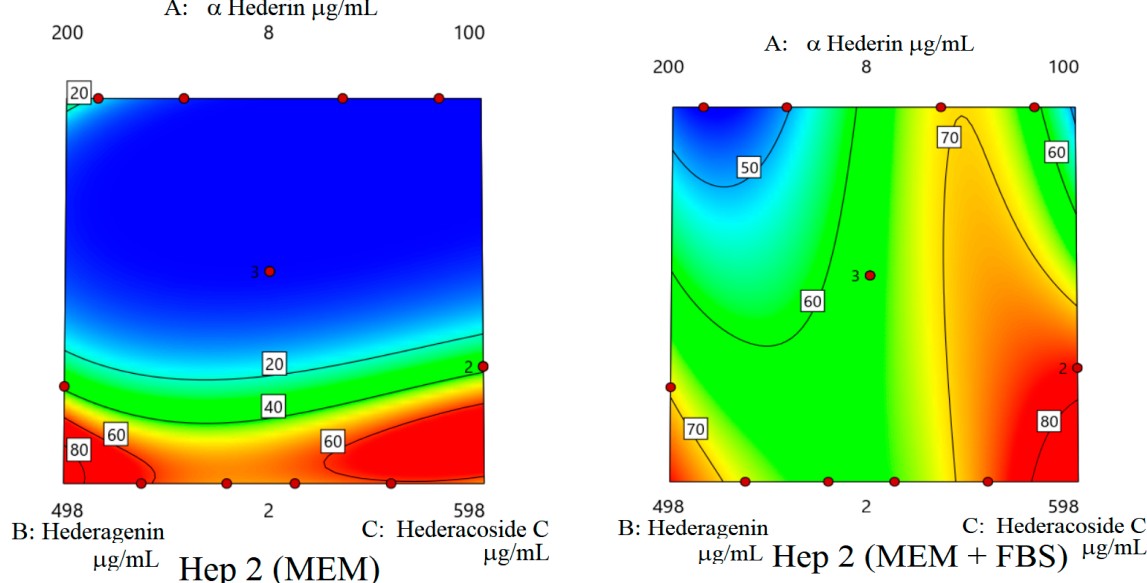

**Figure 4.** Response surfaces for cell viability of Hep-2 (MEM) and Hep-2 (MEM + FBS), concentration zone.

For the combined optimization of response surfaces, we imposed the restrictions presented in Table 7. The solution for these restrictions is presented in Table 8 and the response area in Figure 5.

**Table 7.** Constraints for the response surfaces.

| Name | Goal | Lower Limit | Upper Limit |
|------|------|-------------|-------------|
| A: Hederin | is in range | 2 | 8 |
| B: Hederagenin | is in range | 100 | 498 |
| C: Hederacoside C | is in range | 200 | 598 |
| NCTC | maximize | 79.0 | 80.8879 |
| Hep-2 (MEM) | minimize | 1.614 | 50.0 |
| Hep-2 (MEM + FBS) | none | 44.3192 | 76.9953 |

**Table 8.** Solutions.

| No. | α-Hederin | Hederagenin | Hederacoside C | NCTC | Hep-2 (MEM) | Hep-2 (MEM + SFB) | Desirability |
|-----|-----------|-------------|----------------|------|-------------|-------------------|--------------|
| 1 | 3.863 μg/mL | 100.000 μg/mL | 596.137 μg/mL | 80% | 35% | 76% | 0.182 |

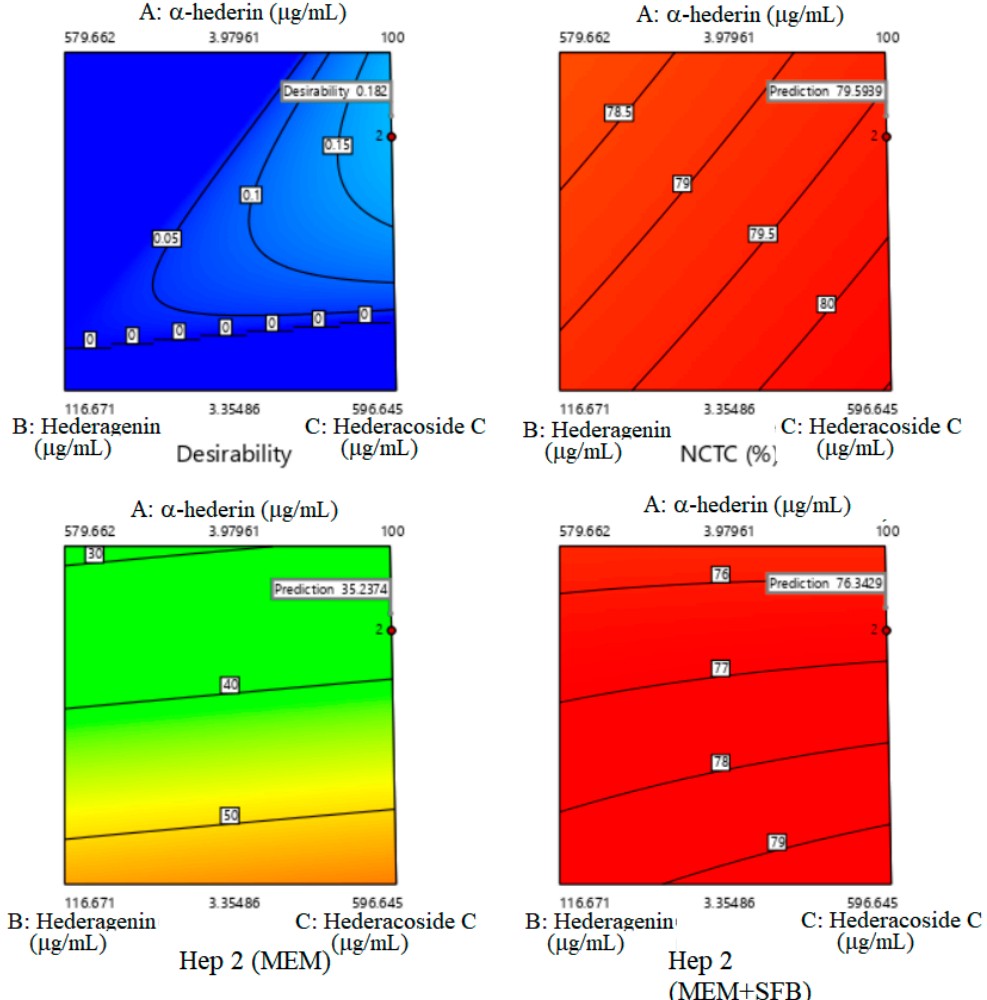

**Figure 5.** The desirability response surface and cell viability for NCTC andHep-2 treated cells.

Figure 5 presents the response area forthe experimental desirability and also for the cell viability of NCTC and Hep-2 treated cells. It maybe seen that there was a fairly restricted area around the experimental point 2, where the two conditions of cell viability were fulfilled, respectively: >79% were met for NCTC cells and <50% for Hep-2 cells. This area had the following coordinates: α-hederin ranging between 3.44 and 3.96; hederagenin ranging from 100 to 116, and hederacoside C between 580 and 595.

A statistical analysis of cell viability data was performed, showing that the finded solution with optimized ratio between the three saponins, α-hederin, hederagenin, and hederacoside C, was the same variant found by restriction solution, in which the 3.863:100.000:596.137 (w/w/w) variant had induced cell viability results of 80% in normal NCTC cells, 35% in Hep-2 tumor cells cultivated in MEM, and 76% in Hep-2 tumor cells in MEM with FBS (Table 8).

From the in vitro cytotoxicity experiments of the 14 variants obtained by mixture modeling using the Design Expert 11 software, the optimal variant of the saponin mixture, i.e., α-hederin, hederagenin, and hederacoside C, (no.9, Table 4), induced similar values: 81% in normal NCTC cells, 36% in Hep-2 tumor cells cultivated in MEM, and 77% in Hep-2 tumor cells in MEM with FBS (Table 4). This saponin combination was biocompatible in NCTC fibroblast cells and had a strong antiproliferative activity in Hep-2 cells in MEM. In addition, it had a mild cytotoxic effect on Hep-2 tumor cells in MEM with FBS. All these data correspond to the response optimization procedures using the desirability function.

## 4. Conclusions

α-Hederin, investigated in Hep-2 epithelial cervix tumor cells cultivated in medium without FBS, presented strong antiproliferative activity (0.5–51% viability) after 24 h of treatment within the entire range of concentrations, i.e., 2–400 μg/mL. Hederagenin exerted moderate antiproliferative activity between 25–400, decreasing the cell viability to 52%. In turn, hederacoside C did not present antiproliferative activity within the entire range of concentrations (2–400 μg/mL). The optimal ratio between the tested saponins was achieved by applying the optimization method with the Design Expert 11 software. The optimized ratio of 3.863:100.000:596.137 (w/w/w) between α-hederin, hederagenin, and hederacoside C was found to be biocompatible for normal NCTC cells (80% viability) and cytotoxic for Hep-2 epithelial cervix tumor cells cultivated in MEM without FBS (35% viability),as confirmed by *in vitro* experiments.

These results provide promising baseline information for the potential use of saponins and mixtures of saponins with antiproliferative effects in the treatment of cancer.

**Author Contributions:** R.T. performed the experiments *in vitro* and wrote the article; C.Z. edited structure formulas, coordinated scientific research, provided critical revision of the article; O.C. and L.M. designed *in vitro* experimental models for bioactive compounds testing, analysed the obtained data and wrote the article; A.O. made the final verification and facilitated the financing of the article. I.C. performed modelling and processing of data with Design Expert 11 and wrote the article. All authors contributed significantly to this work and are in agreement with the content of the manuscript. All authors read the manuscript and approved the final version.

**Funding:** This work was supported by National Project BIODIVERS 25N/2019, 19270102.

**Conflicts of Interest:** The authors confirm that this article content has no conflicts of interest.

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
