# Peer review of "Optimization of Triterpene Saponins Mixture with Antiproliferative Activity"

_applsci, doi:10.3390/app9235160_

Round 1

Reviewer 1 Report

The correction indicated below:

α-hederin, hederagenin and hederacoside C at purity ≥ 98%, Sigma Aldrich  standard saponins are extracted and isolated from ivy?  They are all soluble with the solvent used in the experiments?

What is the software used for statistical analysis for the first part of the article?

Legend is not clear in figure 2. 

The experiment demonstrates a hight cytotoxicity for NCTC in moderate concentrations!

The conclusion is not convincing, the Design Expert 11 software is reliable? Why hederacoside C is not cytotoxic? what are your suggestions? 

The Authors should familiarize themselves with the proper format for references and make appropriate corrections.

Some minor corrections:

Ligne 23: concentrations in 

ligne 43: bioassay

ligne 72: software

ligne 81: minimum 

ligne 94: filters

ligne 99: determined 

ligne 113: measured 

ligne 179: and strong cytotoxicity 

ligne 197: bovine 

ligne 326: moderate 

ligne 334: these results

The manuscript can be accepted after minor revision.

Author Response

Dear Editor,

Thank you for your meaningful comments to manuscript no. applsci-642971.

This is our response to the comments of the reviewers, point by point. The changes are highlighted in red within the revised manuscript. The suggestions made us revised the title of the manuscript to better describe its content.

The correction is indicated below:

α-Hederin, hederagenin and hederacoside C at purity ≥ 98%, Sigma Aldrich standard saponins are extracted and isolated from ivy?  They are all soluble with the solvent used in the experiments? The Sigma Aldrich standard saponins α-Hederin at purity ≥ 90%, hederagenin at purity ≥ 98% and hederacoside C, have no indications about the source used for their obtaining. Yes, they are all soluble with the solvent used in the experiments, respectively dimethyl sulfoxide DMSO.

2.What is the software used for statistical analysis for the first part of the article?

The software used for statistical analysis for the first part of the article is Excel 2007. Legend is not clear in figure 2.

We revised the figure 2 caption in order to clarify the legend: “In vitro cytotoxicity of standard saponins (α-hederin, hederagenin and hederacoside C) cultivated in fibroblasts cells (NCTC) and tumoral cells (Hep-2) cultivated in MEM culture medium or MEM supplemented with FBS (MEM+FBS), determinated by MTT assay. The values represent mean ±SD (n=3).” The experiment demonstrates a high cytotoxicity for NCTC in moderate concentrations! The cytotoxicity assay results of the saponins in NCTC cells showed that α-hederin was biocompatible only until 10 μg/mL, also hederagenin in the range 2 - 100μg/mL, while hederacoside C was biocompatible on the entire studied concentration range 2-400 μg/mL. Using statically Response Surface Method (RSM) analysis with Design Expert 11 software, we aimed to obtain an optimal saponins mixture with maximum antiproliferative effect on Hep-2 tumor cells and biocompatible with NCTC fibroblast cells. The conclusion is not convincing, the Design Expert 11 software is reliable? Why hederacoside C is not cytotoxic? what are your suggestions?

Design Expert 11 it is a software which can be used for experimental design and improvement of products and processes. The RSM by Design Expert 11 software, was used to analyze the cell viability data and generated the optimal ratio between the three saponins α-hederin: hederagenin: hederacoside c, respectively 3.863:100.000:596.137 (w/w/w). The founded variant fulfilled restriction conditions, respectively: > 79% for NCTC cells and < 50% for Hep-2 cells.

The hederacoside C non cytotoxic effect is explained by its chemical structure. Antitumor activity of saponins is influenced by the different characteristics of sugar side chain (sugar linkage association, their number, lipophilicity, or different kinds of carbohydrate groups), which play important roles in their effect. The sugar linkage with the same aglycone and length of sugar chain, determines the antitumor potency [15]. This point is clearly demonstrated by the disaccharide congeners 1→3 linkage that had much lower activity than 1→2 and 1→4 linkages, respectively. The saponins containing the same aglycone in their structure, a small number of glycoside bonds and a reduced length of the carbohydrate chain determine a more powerful anti-tumor effect. From the 3 studied saponins, hederacoside C has more sugar chains in its structure, in consequently the antiproliferative effect is the smallest.

The Authors should familiarize themselves with the proper format for references and make appropriate corrections.

We have revised the references according to the journal’s format.

Some minor corrections:

Ligne 23: concentrations in 

ligne 43: bioassay

ligne 72: software

ligne 81: minimum 

ligne 94: filters

ligne 99: determined 

ligne 113: measured 

ligne 179: and strong cytotoxicity 

ligne 197: bovine 

ligne 326: moderate 

ligne 334: these results

The minor manuscript corrections were made.

Looking forward for your answer.

Best regards,

Assoc. Prof. Ph. D Christina Zalaru

University of Bucharest

Faculty of Chemistry

Department of Organic Chemistry, Biochemistry and Catalysis

Bucharest, Romania

Reviewer 2 Report

The study showed potential effects of saponins on normal fibroblasts cells and cervix ephitelial tumor cells. The study also showed the potential of saponin mixtures in exerting antiproliferative activity.

Could the author provide antiproliferative activity data experimental of mixture saponin on Hep-2 cells and NCTC cells, and compare with separated saponin about their activity?

Express data on IC50 values

Wyhy authors choose to use the three compounds: α-Hederin, Hederagenin and Hederacoside C for study? Are they main compounds in the Hedera helix L.?

Could author provide the in vivo data on tumor of mixture saponin to confirm the your stament about recommend their use in the treatment of cancer disease.

Author Response

Dear Editor,

Thank you for your meaningful comments to manuscript no. applsci-642971.

This is our response to the comments of the reviewers, point by point. The changes are highlighted in red within the revised manuscript. The suggestions made us revised the title of the manuscript to better describe its content.

1. The study showed potential effects of saponins on normal fibroblasts cells and cervix ephitelial tumor cells. The study also showed the potential of saponin mixtures in exerting antiproliferative activity.

1. The study has a part developed in cell culture lab which provided real data on cytotoxicity and antiproliferative activity of saponins (Figure 2) and saponins mixture (Table 4). Another part is the mathematical model (Figures 3 and 4, Tables 5 and 6). Both data types were compared and commented.

2. Could the author provide antiproliferative activity data experimental of mixture saponin on Hep-2 cells and NCTC cells, and compare with separated saponin about their activity?

2. The optimization of saponis mixture provided with RSM by modeling design, generated variant no. 9, the mixture with optimal ratio between the three saponins α-hederin: hederagenin: hederacoside C, respectively 3.863:100.000:596.137(w/w/w). This saponins mixture was tested in cell culture and exerted maximum antiproliferative effect in Hep-2 tumor cells in MEM without FBS (35.52% cell viability) and Hep-2 tumor cells cultivated in MEM (76.53%) and in the also was biocompatible with NCTC fibroblast cells (80.89% viability). The comparison between cytotoxicity results from cell culture and data modelling for the optimal mixture of 3 saponins indicated a perfect match.

3. Express data on IC50 values

3. For some compounds they were too high to be determined from the range of tested concentrations.

4. Why authors choose to use the three compounds: α-Hederin, Hederagenin and Hederacoside C for study? Are they main compounds in the Hedera helix L.?

4. The 3 saponins α-hederin, hederagenin and hederacoside C are main compounds in the Hedera helix L.

5. Could author provide the in vivo data on tumor of mixture saponin to confirm the your stament about recommend their use in the treatment of cancer disease.

5. At present stage our results provided baseline information for the saponin mixtures with antiproliferative effect, and their potential to be used for pharmacological applications in the treatment of cancer disease. We are intending to continue our study for obtaining further in vivo data to recommend their use in the treatment of cancer disease.

Looking forward for your answer.

Best regards,

Assoc. Prof. Ph. D Christina Zalaru

University of Bucharest

Faculty of Chemistry

Department of Organic Chemistry, Biochemistry and Catalysis

Bucharest, Romania

Reviewer 3 Report

This study presented the cytotoxic effects of α-hederin, hederagenin and hederacoside C isolated from  Hedera helix L. They also performed an analysis of the optimal mixture composed of these three ingredients and test their cytotoxic effects. I couldn't support the publication of the current version based on the following reasons.

Too few data were presented. Only one cancer cell line was used in this study.  It appears to me that all the antiproliferative effects (as shown by cell viability) are due to α-hederin whose anticancer effects have been widely studied.  The cytotoxic effects of cocultured without FBS is a lack of scientific significance. This study failed to provide a clear conclusion and direction for further research and application. 

Author Response

Dear Editor,

Thank you for your meaningful comments to manuscript no. applsci-642971.

This is our response to the comments of the reviewers, point by point. The changes are highlighted in red within the revised manuscript. The suggestions made us revised the title of the manuscript to better describe its content.

This study presented the cytotoxic effects of α-hederin, hederagenin and hederacoside C isolated from Hedera helix L. They also performed an analysis of the optimal mixture composed of these three ingredients and test their cytotoxic effects. I couldn't support the publication of the current version based on the following reasons.

Too few data were presented. Only one cancer cell line was used in this study. It appears to me that all the antiproliferative effects (as shown by cell viability) are due to α-hederin whose anticancer effects have been widely studied.  The cytotoxic effects of cocultured without FBS is a lack of scientific significance. This study failed to provide a clear conclusion and direction for further research and application.

The aim of this study was to evaluate the in vitro cytotoxic and antiproliferative activity induced by the three triterpenic saponins, α-hederin, hederagenin and hederacoside C and to optimize a mixture of the three standard saponins for a maximal antiproliferative effect. For this purpose, the three standard saponins were tested for cytotoxicity in normal mouse fibroblasts (NCTC cells) and antiproliferative activity in human epithelial cervix tumor cells (Hep-2). Antiproliferative effect was exerted by a-hederin but also by hederagenin. The obtained experimental results were analyzed by Response Surface Method (RSM) provided by Design Expert 11 software [28], correlating the concentration of studied saponins with the antiproliferative activity. A mixture design model using Design Expert 11 software was proposed and tested. Testing in medium without FBS was performed in order to avoid the interaction with proteins albumin, globulin, gamma globulin presented in FBS as previously showed [8, 32]. The conclusions and further research, but also the entire manuscript were revised and reformulated.

Looking forward for your answer.

Best regards,

Assoc. Prof. Ph. D Christina Zalaru

University of Bucharest

Faculty of Chemistry

Department of Organic Chemistry, Biochemistry and Catalysis

Bucharest, Romaniav

Round 2

Reviewer 3 Report

Thanks for the improved revision.